

# Baseline Surface Radiation Network (BSRN): structure and data description (1992-2017)

Amelie Driemel[1], John Augustine[2], Klaus Behrens[3,27], Sergio Colle[4], Christopher Cox[5], Emilio Cuevas-Agulló[6], Fred M Denn[7], Thierry Duprat[8], Masato Fukuda[9], Hannes Grobe[1], Martial Haeffelin[10], Nicole Hyett[11], Osamu Ijima[9], Ain Kallis[12], Wouter Knap[13], Vasilii Kustov[14], Charles N Long[2], David Longenecker[2,27], Angelo Lupi[15], Marion Maturilli[16], Mohamed Mimouni[17,27], Lucky Ntsangwane[18], Hiroyuki Ogihara[9], Xabier Olano[19], Marc Olefs[20], Masao Omori[9], Lance Passamani[11], Enio Bueno Pereira[21], Holger Schmithüsen[1], Stefanie Schumacher[1], Rainer Sieger[1,28], Jonathan Tamlyn[22,27], Roland Vogt[23], Laurent Vuilleumier[24], Xiangao Xia[25], Atsumu Ohmura[26,27], and Gert König-Langlo[1,27]

[1]Alfred-Wegener-Institut Helmholtz-Zentrum für Polar- und Meeresforschung, Bremerhaven, Germany
[2]NOAA ESRL Global Monitoring Division, Boulder, CO, USA
[3]Deutscher Wetterdienst, Meteorologisches Observatorium Lindenberg - Richard-Aßmann-Observatorium, Germany
[4]Universidade Federal de Santa Catarina, Florianopolis, Brasil
[5]CIRES/NOAA-ESRL Physical Sciences Division, Boulder, CO, USA
[6]Izaña Atmospheric Research Center (AEMET), Tenerife, Spain
[7]Science Systems and Applications, Inc., Lanham, USA
[8]Meteo France, Carpentras, France
[9]Japan Meteorological Agency, Tokio, Japan
[10]Laboratoire de Météorologie Dynamique, Paris, France
[11]Bureau of Meteorology, Melbourne, Australia
[12]Estonian Environment Agency, Toravere, Estonia
[13]Royal Netherlands Meteorological Institute, De Bilt, Netherlands
[14]Arctic and Antarctic Research Institute, Saint Petersburg, Russia
[15]Institute of Atmospheric Sciences and Climate, National Research Council of Italy, Bologna, Italy
[16]Alfred-Wegener-Institut Helmholtz-Zentrum für Polar- und Meeresforschung, Potsdam, Germany
[17]Office National de la Météorologie, Algier, Algeria
[18]South African Weather Service, Pretoria, South Africa
[19]National Renewable Energy Centre, Sarriguren, Spain
[20]ZAMG - Zentralanstalt für Meteorologie und Geodynamik, Vienna, Austria
[21]Instituto Nacional de Pesquisas Espaciais, São José dos Campos, Brasil
[22]Met Office, Exeter, Devon, United Kingdom
[23]Meteorology Climatology and Remote Sensing, Department Environmental Sciences, University of Basel, Basel, Switzerland
[24]Federal Office of Meteorology and Climatology, MeteoSwiss, Payerne, Switzerland
[25]LAGEO, Institute of Atmospheric Physics, Chinese Academy of Sciences, Beijing, China
[26]Institute for Atmospheric and Climate Science, ETH Zurich, Zurich, Switzerland
[27]retired
[28]deceased

**Correspondence:** Amelie Driemel (amelie.driemel@awi.de)



**Abstract.** Small changes in the radiation budget at the earth's surface can lead to large climatological responses when persistent over time. With the increasing debate on anthropogenic influences on climatic processes during the 1980s the need for accurate radiometric measurements with higher temporal resolution was identified, and it was determined that the existing measurement networks did not have the resolution or accuracy required to meet this need. In 1988 the WMO therefore proposed the estab-

lishment of a new international Baseline Surface Radiation Network (BSRN), which should collect and centrally archive high quality ground-based radiation measurements in 1-minute resolution. BSRN began its work in 1992 with 9 stations, currently (status 2018-01-01), the network comprises 59 stations (with data) and 9 candidates (stations recently accepted into the network with data forthcoming to the archive) distributed over all continents. The BSRN database is the World Radiation Monitoring Center. It is hosted at the Alfred Wegener Institute in Bremerhaven, Germany and now offers more than 10300 months of data

from the years 1992 to 2017. All data are available at https://doi.pangaea.de/10.1594/PANGAEA.880000 free of charge.

## 1 Introduction

The climate of our earth is decisively influenced by radiative processes. Even small changes in the radiation budget at the earth's surface can lead to large climatological responses when persistent over time (Chylek et al., 2007; Kwok and Untersteiner, 2011). International data centers have long been archiving data that is relevant for studies aiming to detect and monitor long-term

variability in the surface energy balance. For example, the Global Energy Balance Archive (GEBA) (http://www.geba.ethz.ch/) has archived monthly means of various energy flux components from around 2500 locations since 1985 (Wild et al., 2013, 2017). Similarly, the "World Radiation Data Center" (http://wrdc.mgo.rssi.ru/) under the auspices of the World Meteorological Organization (WMO), has provided daily and monthly sums and monthly means of solar radiation fluxes since 1964 from more than 1000 stations. A small subset of these stations that are connected to the Global Atmosphere Watch (GAW) also provide

hourly sums.

However, with the increasing debate on anthropogenic influences on climatic processes since the 1980s - e.g. anthropogenic aerosol emission (Charlson et al., 1992) - the need for accurate radiometric measurements with higher temporal resolution was identified, and it was determined that the existing measurement networks could not provide the necessary resolution or accuracy. As a solution, in 1988 the WMO proposed the establishment of a new international Baseline Surface Radiation

Network (BSRN) under the World Climate Research Program (WCRP). The BSRN would provide high-temporal-resolution ground-based radiation measurements for the validation of satellite data, the validation and improvement of radiative transfer calculations in climate models, and support the detection and monitoring of long-term changes in ground-level radiation fluxes. The requirements of BSRN stations that set the network apart from its predecessors include the following (Ohmura et al., 1998; McArthur, 2005, see also): a commitment for long-term involvement of the station under the lead of a radiation expert respon-

sible for the quality of the data; measurements of direct, diffuse and global downwelling shortwave fluxes and the downwelling longwave flux in one minute resolution; and traceable calibrations of the radiation instruments to the World Radiometric Reference (WRR), which is maintained at the World Radiation Center (WRC) in Davos, Switzerland. Furthermore, stations should be representative of a relatively large surrounding area for use in satellite validation, and should be manned at least during work



days to allow for regular cleaning/maintenance/check of instruments. To guarantee high quality measurements, ventilation of all radiation instruments, especially at sites with large variability in ambient temperature, should be installed.

A central data archive for BSRN was developed under the direction of Atsumu Ohmura at the Swiss Federal Institute of Technology (ETH) in Zurich (Gilgen et al., 1995; Hegner et al., 1998), called the World Radiation Monitoring Center (WRMC).

Since 2008, the WRMC has been run by the Alfred Wegener Institute (AWI), Helmholtz Center for Polar and Marine Research in Bremerhaven, Germany (König-Langlo et al., 2013) and currently stores more than 10300 months of radiometric data at one-minute temporal resolution. In this manuscript, we describe the structure of the BSRN and the WRMC, the data coverage, data quality procedures, and we provide information for potential users interested in accessing BSRN data from 1992-2017.

## 2   BSRN structure and measured values

BSRN began its work in 1992 with 9 stations distributed globally, including the Antarctic, Arctic, Atlantic and Pacific Oceans, North America, Africa and Europe. Currently (status 2018-01-01), the network comprises 59 stations (with data) and 9 candidates (stations recently accepted into the network with data forthcoming to the archive), collectively representing all seven continents as well as island-based stations in the Pacific, Atlantic, Indian and Arctic Oceans (Figure 1). Each BSRN station is committed to measuring at least the following radiation fluxes (SI unit W m-2) continuously in high temporal resolution (about

1 Hz) and providing quality-tested means for one-minute intervals:

- **Global short wave radiation (SWD)**: The incoming solar radiation flux incident upon the planet's surface. Downward solar radiation, downward short-wave radiation or global horizontal irradiance are synonyms. SWD is measured with pyranometers, which typically cover a spectral range between 250 and 3000 nm.

- **Direct radiation (DIR)**: The part of the global radiation incident on a surface orthogonal (or "normal") to the sun's
beam, which originates from the solid angle subtended by the sun's disk. It is also called direct normal irradiance. Most BSRN stations use pyrheliometers which are automatically aligned perpendicular to the sun's beams by means of sun trackers.

- **Diffuse radiation (DIF)**: The component of global radiation that is scattered out of the solar beam by atmospheric constituents. It is also called diffuse horizontal irradiance. DIF is measured with pyranometers mounted horizontally that
are shaded, e.g., by shadows cast by discs or balls, also using a sun tracker.

- **Long-wave downward radiation (LWD)**: This is the thermal emission of the atmosphere incident on the planet's surface. The term thermal irradiation is also used. LWD is measured with pyrgeometers that typically cover a spectral range between 4000 and 40,000 nm. Pyrgeometers are also frequently shaded to help mitigate solar leakage and help maintain a consistent temperature between the sensor window and the instrument case.

Twenty-three BSRN stations (Table 1, Figure 1) additionally provide the following upward directed radiation fluxes, allowing for the calculation of net surface radiation:



- **Short-wave upward radiation (SWU)**: The part of the global radiation that is reflected by the surface.

- **Long-wave upward radiation (LWU)**: The thermal emission from the planet's surface.

Apart from the measurements mentioned above, many stations also offer additional information such as spectral ultraviolet irradiance, synoptic weather observations, radiosonde data (vertical profiles of temperature, humidity, wind and ozone), as well as data on the ozone column thickness. An up to date overview of all available measurements can be accessed here: https://dataportals.pangaea.de/bsrn/, an explanation of the content of each logical record (LR) can be found in Table 1. These additional measurements are not required for BSRN stations (though they are recommended) and in this manuscript we focus on the core measurements of BSRN, the downward- and upward-directed long- and short-wave radiation fluxes from 1992 to 2017.

## 3    Data format and quality checks

### 3.1    Data format

The original submission format is the so-called "station-to-archive-file". The relatively complex, strictly defined format of the station-to-archive file is intended to encapsulate both the data and all relevant metadata within a single ASCII file. Metadata records include the contact information, calibration history, geographical characteristics of the station as well as the models of the instruments. The actual data within station-to-archive files are divided into so-called logical records (LR), which do not contain table headers (no parameter names/units); see König-Langlo et al. (2013) for a detailed explanation of the content of each logical record and Table 2 in the present manuscript for an overview. All station-to-archive-files are stored on, and are accessible via, a public ftp server. To increase usability and visibility of the files, the WRMC at the AWI also publishes the data within the "Data Publisher for Earth & Environmental Science" PANGAEA (http://bsrn.awi.de/data/data-retrieval-via-pangaea/) in a much more straightforward and intuitive format. BSRN datasets in PANGAEA contain a metadata header followed by the data table with parameter names and units, see e.g. https://doi.org/10.1594/PANGAEA.874567 (König-Langlo, 2017), and Figure 2 for a visualisation of this data file.

### 3.2    Quality control

The methods for the initial stage of quality control are left up to the individual station scientists and thus are not implemented uniformly across BSRN stations. However, BSRN does maintain recommendations (Long and Dutton, 2002; Long and Shi, 2008) as well as software tools (Schmithüsen et al., 2012). After this initial quality control, the data generated at BSRN stations are submitted in monthly granularity in the station-to-archive file format by the station scientist to the WRMC data curator. From 1992 to 2007, when the archive was maintained by the ETH in Zurich, the data were checked for physically possible limits within the WRMC and suspicious values were flagged, but not removed. Since being operated at the AWI, additional procedures have been enacted for data submitted to the WRMC. The data are now not only quality checked (see below), but suspicious data are reported back to the station scientist with an appeal to remove them and resubmit the file. The files that were



archived using the old system (pre-2007) were adopted by AWI without further checks and without quality flags. However, many of the old files have since been exchanged with updated versions. Details on the quality checks performed at the AWI are given below.

Station scientists and users have the option of processing BSRN data using software developed for the BSRN community and WRMC called the "BSRN Toolbox" (Schmithüsen et al., 2012). The software provides a format check for the station-to-archive files (and for PANGAEA download files, see below) and can perform quality checks of the data as outlined in the "BSRN Global Network Recommended QC tests, V2.0" (Long and Dutton, 2002). The types of tests include those that identify data outside physically possible limits, as well as extremely rare limits. Additionally, a set of comparison metrics are available that flag data as suspect based on empirically established relationships relative to other variables ("Comparisons" check, e.g. LWD to air temperature). A broader and more refined range of such tests are outlined by Long and Shi (2008) and their implementation is recommended for BSRN. The astronomical data necessary for many tests can be calculated using different algorithms (Iqbal 1983, Solpos with refraction and Solpos without refraction, Michalsky), most of them selectable within the BSRN Toolbox. At the AWI the data curator generally uses the Iqbal 1983 algorithm and the "Comparisons" check. In all cases, an output file can be created which includes the quality codes (checkbox "Quality codes") and another file which plots the measured global radiation (SWD) vs. the calculated "total horizontal" radiation (SumSW), computed from the direct and diffuse radiation measurements (SumSW = DIF + DIR * cos(z); see Figure 3 for details). A comparison of SWD to SumSW is a good way of detecting instrument malfunctions[1].

Since the BSRN Toolbox is subject to the GNU General Public License, it is freely available to everyone. With the help of the Toolbox, BSRN station scientists as well as data users can easily download and convert station-to-archive files to user-friendly formats by clicking the option "Data", which splits station-to-archive files up into logical records containing parameter names and units. In addition, quality checks can be applied on BSRN files (downloaded from both, ftp and PANGAEA) using the option "Quality check". All data can be visualized with the free software PanPlot2 (Sieger and Grobe, 2013).

Some quality issues cannot be easily solved, though. One prominent example is the case of zero-offsets: Thermophile radiometers are known to suffer a negative bias due to infrared loss to the sky (Dutton et al., 2001). Consequently, though physically impossible, negative radiation fluxes/values are often present in nighttime measurements. To reduce these so-called "IR loss offsets", BSRN sensors are ventilated (Michalsky et al., 2017) and/or heated artificially. However, ventilation sometimes fails or is insufficient to compensate large temperature variations. Depending on the location and instrument used, negative nighttime values >-4 W m-2 can just be removed, or in other cases corrections can be made using algorithms that make use of the other co-located data that is part of the BSRN instrument suite (Long et al., 2001; Younkin and Long, 2004). However, if the negative bias also occurs - but is not visible - during the day, the BSRN recommends keeping the nighttime negatives in the archived files. Each individual BSRN customer can then decide how to treat the data depending on the scope and aim of their work. Ventilation is also used to help prevent the build-up of ice (e.g., rime or frost) on the sensor windows, though experience

---

[1]although if the sun tracker for DIR and DIF is completely off then DIR = 0 and SumSW = SWD (very good agreement), so there are situations where a perfect match does not indicate high quality data. But this test is non-definitive in that failure does not indicate which of the three instruments might be producing the disagreement



within the BSRN community indicates that ice remains an important vulnerability to measurement quality and data retention rates for stations in cold climates (BSRN, 2012).

A more thorough description of BSRN methods and quality control issues can be found in Lanconelli et al. (2011) and Matsui et al. (2012), where issues especially related to polar BSRN sites are elaborated on. Other interesting papers on the
quality and possible biases of radiation measurements were published by Vuilleumier et al. (2014), Olefs et al. (2016) and Nyeki et al. (2017).

## 4 Data availability

In https://doi.pangaea.de/10.1594/PANGAEA.880000 the downward and upward short- and long-wave data of all BSRN stations for the years 1992 to 2017 are archived. More than 10300 monthly files (>850 station years) of high quality radiation data
are thus available free of charge. The files are organized in zip files with one file containing all monthly data files for one year of downward and (if available) upward radiation data (LR0100+0300 or LR0100 and LR3010) from one station. The format of the files is tab delimited txt, and the structure is the same as if downloaded directly from PANGAEA (see Figure 4). At the top of the files the metaheader gives details on the citation of the dataset, a link to the respective station-to-archive file on the ftp server, the location of the station, parameters, units, PI, methods etc., followed by the actual data table with abbreviated
parameter names and units.

To be able to download the files, users should contact the WRMC staff (http://bsrn.awi.de/contact-persons/) to request a BSRN account. By accepting this account, the data user agrees to comply with the guidelines defined at http://bsrn.awi.de/data/conditions-of-data-release/. The reason for requiring that users register with BSRN is that the organization is a non-profit voluntary network and it is important that station managers who commit resources to providing the data are able to receive
appropriate acknowledgement, the correct citation of their data files in publications and are able to monitor level of use of the station data. Therefore, the acknowledgement of the data release guidelines is an essential step in the data download process. There are plans to simplify the procedure by allowing the acceptance of the data release guidelines directly during the download of the data, but this feature is not operative yet. From 2017-01-01 to 2017-12-31 140 requests from 35 countries were received and answered promptly (within one day). On average 800 visitors per month visit the BSRN website. Up to now, Web of
Science reports a total of 100 publications related to BSRN which have been cited more than 2500 times (accessed on 2018-01-10). Several hundred to several thousand BSRN files are downloaded each month. Unfortunately, to the best knowledge of the authors there is no data citation report yet available to check for BSRN dataset citations.

## 5 Some remarks for the best usage of BSRN data

The zip files provided in https://doi.pangaea.de/10.1594/PANGAEA.880000 contain all downward and (where available) up-
ward radiation fluxes of one station and one year in monthly granularity. The ".tab" files can be opened in any text editor or spreadsheet program (e.g. Excel). They can be visualized individually (in the provided monthly granularity) e.g. in PanPlot2





(see Section 3.2) by dragging the file onto the program icon. To obtain annual files one can e.g. drag all twelve months on the BSRN Toolbox, and go to "Tools" => "Concatenate files". One can then insert how many lines the program should skip (metadata header) and press OK. Be aware that the resulting annual file will be quite large.

PANGAEA itself offers comfortable ways to extract long periods of individual variables via the so-called "data warehouse" (http://bsrn.awi.de/data/data-retrieval-via-pangaea/data-warehouse/). Go to https://dataportals.pangaea.de/bsrn/, choose the station, logical record and time period and click on the respective value or 'X' (for all datasets from that year and station) which will then lead to the respective search page on www.pangaea.de. Before the user can retrieve the selected files he/she has to log in at the very top right corner of this page (log in can be obtained from the WRMC director, see http://bsrn.awi.de/contact-persons/). After logging in, a green button "Data Warehouse" appears on the top right corner of the page. A click on this button leads into the Data Warehouse where parameter(s) can be chosen individually. Arithmetic averages can be calculated via "Method" but the results should be considered with care in cases where significant data gaps exist. In these cases the user should refer to more sophisticated averaging procedures as published in Roesch et al. (2011) or using a macro excel spreadsheet for hourly averages provided by Schild (2016). After clicking on "Start Data Warehouse Query" a window will pop up to download the output-file. Depending on the amount of years, stations or parameters chosen the files can get quite large (easily >100 MB).

A list of all radiation instruments used in BSRN is available at https://www.pangaea.de/ddi?request=bsrn/BSRNMethods&format=html&title=BSRN+Methods. To get all data from one instrument just copy-paste the instrument specifications into https://www.pangaea.de/ and press search.

## 6  Summary and outlook

In the 25 years of its existence, the Baseline Surface Radiation Network (BSRN) and its central archive have become an important component of global climate monitoring and climate research. Beginning as a contribution to the "Global Energy and Water Cycle Experiment" (GEWEX), the BSRN was later made part of the "Global Climate Observing System" (GCOS). Also, a formal cooperation agreement was established with the "Network for the Detection of Atmospheric Composition Change" (NDACC).

Validation of models (Wild et al., 2008; Müller Schmied et al., 2016) and satellite-based estimates of surface radiation (Pinker et al., 2005) remains the focus of users. However, the data have also been used for trend analyses (Long et al., 2009) and aerosol studies (Xia et al., 2007; Kim and Ramanathan, 2008). Over the past decade, BSRN data have increasingly been used by customers in the field of solar energy, who especially welcome the high temporal resolution of the data (e.g. Moshövel et al. 2015, Hofmann and Seckmeyer 2017). More reviewed scientific papers referring to BSRN can be found at http://bsrn.awi.de/other/publications/.

Neither the BSRN stations nor the archive receive direct financial grants for their involvement in the BSRN. Despite this fact it has succeeded in developing a network that is setting standards for ground-level radiation measurements through volunteer efforts by a global community of scientists and technicians who have agreed to define and maintain a common set of standards.



Twelve of the 59 stations have unfortunately closed, are no longer maintained, are inactive, or seek funding. Thanks to the BSRN, the data collected by these stations remain available. However, BSRN is now in the process of adding nine new stations into the network (four from India, two from Taiwan, one from Russia, one from the Azores and one from Australia). One of the challenges the BSRN faces will therefore be the tremendous effort to continue its collection of high quality, high

temporal resolution measurements with as many stations as possible. As a fundamental environmental variable, the importance of permanently archived, freely accessible and high quality radiation data cannot be valued highly enough.

*Competing interests.* The authors declare that they have no conflict of interest.

*Acknowledgements.* First and foremost, we would like to express our gratitude to all BSRN scientists and technicians working tediously to keep the high quality of BSRN standards up. Your dedicated work of calibrating, maintaining and quality checking sensors, instruments

and data - often working extra hours to do so - is greatly appreciated. Thanks as well to all the institutions supporting the BSRN cause! We would also like to thank Wolfgang Cohrs for his excellent work in keeping the ftp server up and running, as well as his work for the BSRN homepage. Last but not least we are indebted to the current (Skalde Lübberstedt) and former (Bonnie Raffel, Friedrich Richter) WRMC Data Curators - dedicated students with a knack for data visualization and the mind set on "finding the needle in the data haystack". This article is dedicated to co-author Rainer Sieger, architect of the BSRN data archive in PANGAEA, responsible for maintaining the BSRN Toolbox and

a good friend, who passed away unexpectedly in 2017. And also to Ells Dutton, the original and long-serving BSRN Project Manager for the first two decades, who was essential in establishing BSRN as a leader in surface radiation measurements and improvements. Ells passed away suddenly in 2012.



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



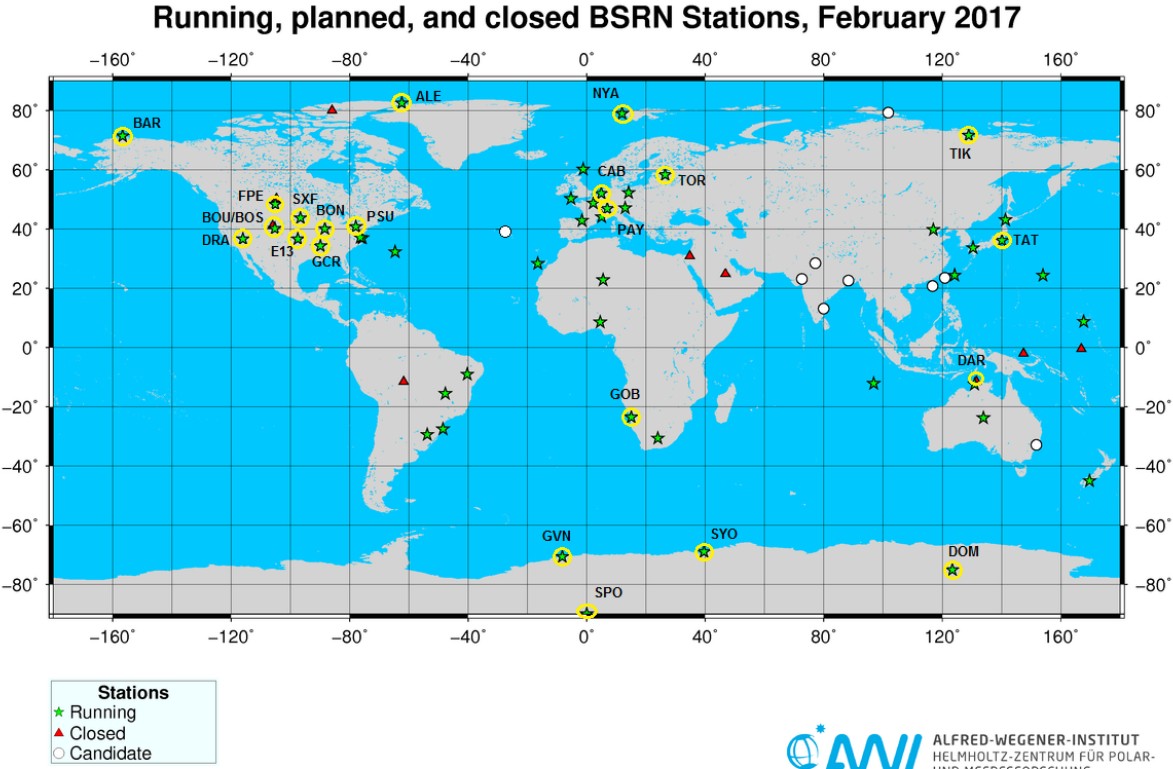

**Figure 1.** Active (green star) and closed (red triangle) BSRN stations. Stations that measure both upward and downward radiative fluxes are circled with yellow and labelled with their BSRN abbreviation. White: BSRN candidate stations.



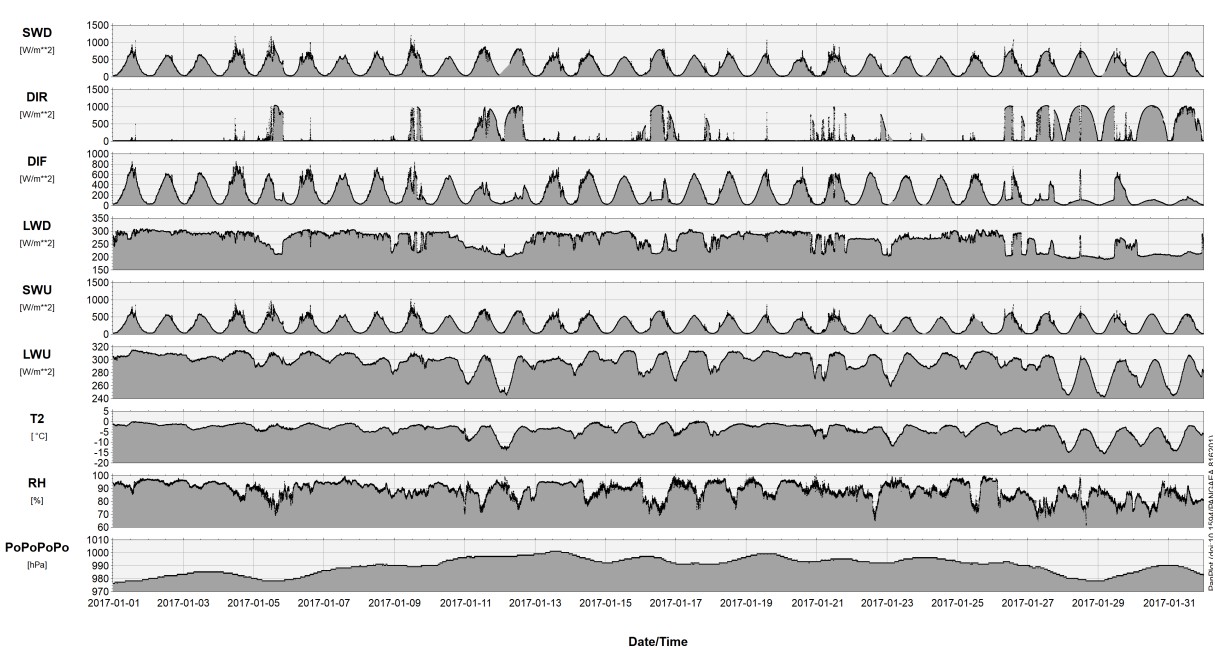

**Figure 2.** Example of a PanPlot2 (Sieger and Grobe, 2013) visualisation of a monthly data file (January 2017) from station GVN (Neumayer, Antarctica); T2 = air temperature at 2 m height, RH = relative humidity, PoPoPoPo = air pressure





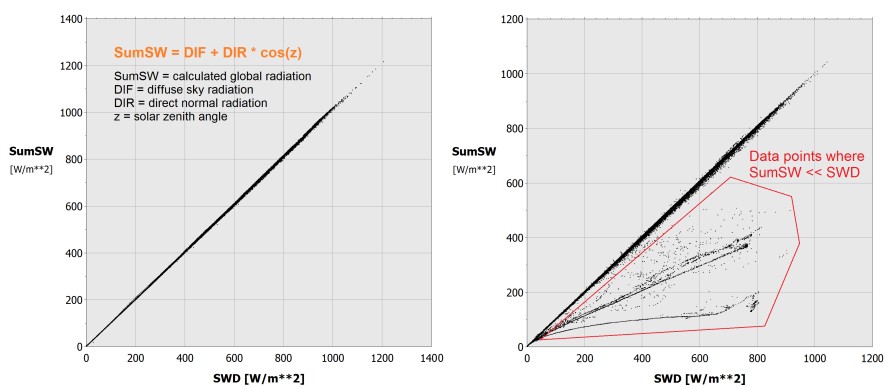

**Figure 3.** Left: Very good agreement of SumSW with SWD indicating high quality measurements of SWD, DIR and DIF (please note that due to the thermal offset, combined with the cosine error of SWD the plot usually exhibits more scatter than shown here). Right: three days of instrument failure shown by data points far below the 1:1 line. The Toolbox flags values where SWD/SumSW is not within +/- 8% of 1.0 for solar zenith angle (SZA) < 75° respectively where SWD/Sum SW is not within +/- 15% of 1.0 for 93° > SZA > 75°. For Sum SW < 50 W/m2 however, the test is not effective.



/* METADATA:

|  | König-Langlo, Gert (2017): Basic and other measurements of radiation at Neumayer Station (2017-01). Alfred Wegener Instit |
|---|---|
| **Citation:** | Bremerhaven, PANGAEA, https://doi.org/10.1594/PANGAEA.874567 |
| **Other version:** | König-Langlo, Gert (2017): BSRN Station-to-archive file for Neumayer Station (2017-01). ftp://ftp.bsrn.awi.de/gvn/gvn0117.d |
| **Project(s):** | Baseline Surface Radiation Network (BSRN) (URI: http://bsrn.awi.de) |
| **Coverage:** | LATITUDE: -70.650000 * LONGITUDE: -8.250000 |
|  | DATE/TIME START: 2017-01-01T00:00:00 * DATE/TIME END: 2017-01-31T23:59:00 |
| **Event(s):** | GVN (Georg von Neumayer) * LATITUDE: -70.650000 * LONGITUDE: -8.250000 * DATE/TIME: 1992-01-01T00:00:00 * ELEVAT |
| **Parameter(s):** | **DATE/TIME (Date/Time)** * GEOCODE * PI: König-Langlo, Gert gert.koenig-langlo@awi.de |
|  | **HEIGHT above ground [m] (Height)** * GEOCODE * PI: König-Langlo, Gert gert.koenig-langlo@awi.de |
|  | **Short-wave downward (GLOBAL) radiation [W/m**2] (SWD)** * PI: König-Langlo, Gert gert.koenig-langlo@awi.de * METHOD |
|  | **Direct radiation [W/m**2] (DIR)** * PI: König-Langlo, Gert gert.koenig-langlo@awi.de * METHOD: Pyrheliometer, Kipp & Zone |
|  | **Diffuse radiation [W/m**2] (DIF)** * PI: König-Langlo, Gert gert.koenig-langlo@awi.de * METHOD: Pyranometer, Kipp & Zone |
|  | **Long-wave downward radiation [W/m**2] (LWD)** * PI: König-Langlo, Gert gert.koenig-langlo@awi.de * METHOD: Pyrgeome |
|  | **Short-wave upward (REFLEX) radiation [W/m**2] (SWU)** * PI: König-Langlo, Gert gert.koenig-langlo@awi.de * METHOD: Py |
|  | **Long-wave upward radiation [W/m**2] (LWU)** * PI: König-Langlo, Gert gert.koenig-langlo@awi.de * METHOD: Pyrgeometer |
|  | **Air temperature at 2 m height [°C] (T2)** * PI: König-Langlo, Gert gert.koenig-langlo@awi.de * METHOD: Thermometer |
|  | **Humidity, relative [%] (RH)** * PI: König-Langlo, Gert gert.koenig-langlo@awi.de * METHOD: Hygrometer |
|  | **Station pressure [hPa] (PoPoPoPo)** * PI: König-Langlo, Gert gert.koenig-langlo@awi.de * METHOD: Barometer |

*/

| Date/Time | Height [m] | SWD [W/m**2] | DIR [W/m**2] | DIF [W/m**2] | LWD [W/m**2] | SWU [W/m**2] | LWU [W/m**2 |
|---|---|---|---|---|---|---|---|
| 2017-01-01T00:00 | 2 | 19 | 0 | 19 | 243 | 18 | 303 |
| 2017-01-01T00:01 | 2 | 19 | 0 | 19 | 243 | 18 | 302 |
| 2017-01-01T00:02 | 2 | 20 | 0 | 19 | 245 | 18 | 302 |
| 2017-01-01T00:03 | 2 | 20 | 0 | 20 | 248 | 18 | 302 |
| 2017-01-01T00:04 | 2 | 20 | 0 | 19 | 246 | 18 | 302 |
| 2017-01-01T00:05 | 2 | 20 | 0 | 19 | 243 | 18 | 302 |
| 2017-01-01T00:06 | 2 | 20 | 0 | 19 | 244 | 18 | 302 |
| 2017-01-01T00:07 | 2 | 20 | 0 | 20 | 245 | 18 | 302 |
| 2017-01-01T00:08 | 2 | 21 | 0 | 20 | 250 | 19 | 302 |
| 2017-01-01T00:09 | 2 | 21 | 0 | 21 | 258 | 19 | 303 |

**Figure 4.** Example of the PANGAEA download format of a monthly data file (January 2017) from station GVN (Neumayer, Antarctica).





Table 1. Overview of the logical records (LR) contained in station-to-archive files, as well as number of stations submitting these data to the WRMC (status December 2017).The last column states the standard title for BSRN datasets archived within PANGAEA ((1) in PANGAEA the LR0300 and LR0100 are archived as one dataset)

| Logical Record | Data type included in the LR | No. Of stations | Dataset title in PANGAEA |
|---|---|---|---|
| LR0100 | Global, Diffuse, Direct, Long-wave down | 59 | Basic measurements of radiation.. |
| LR0300 | Reflected, Long-wave up | 15 | Basic and other measurements . . . (1) |
| LR0500 | Ultraviolet radiation | 12 | Ultra-violet measurements .. |
| LR1000 | Synops | 13 | Meteorological synoptical observations.. |
| LR1100 | Upper air soundings | 30 | Radiosonde measurements .. |
| LR1200 | Total ozone | 9 | Ozone measurements .. |
| LR1300 | Ceilometer data | 3 | Expanded measurements .. |
| LR3010 | Radiation measurements from tower | 13 | Other measurements at 10 m .. |



**Table 2.** BSRN stations and the years where short- and longwave radiation data is available for each station. (1) stations that measure all components necessary to calculate a net radiation budget (i.e. SWD, SWU, LWD, LWU), (2) stations with upward radiation fluxes measured from a tower (10 m height)

| Station | Short name | 1992 | 1993 | 1994 | 1995 | 1996 | 1997 | 1998 | 1999 | 2000 | 2001 | 2002 | 2003 | 2004 | 2005 | 2006 | 2007 | 2008 | 2009 | 2010 | 2011 | 2012 | 2013 | 2014 | 2015 | 2016 | 2017 |
|---|---|---|---|---|---|---|---|---|---|---|---|---|---|---|---|---|---|---|---|---|---|---|---|---|---|---|---|
| Alert | ALE (1) | | | | | | | | | | | | | X | X | X | X | X | X | X | X | X | X | X | | | |
| Alice Springs | ASP | | | | X | X | X | X | X | X | X | X | X | X | X | X | X | X | X | X | X | X | X | X | X | X | |
| Barrow | BAR (1) | X | X | X | X | X | X | X | X | X | X | X | X | X | X | X | X | X | X | X | X | X | X | X | X | X | X |
| Bermuda | BER | X | X | X | X | X | X | X | X | X | X | X | X | X | X | X | X | X | X | X | X | X | X | | | X | X |
| Billings | BIL | | X | X | X | X | X | X | X | X | X | X | X | X | X | X | X | X | X | X | X | X | X | X | | | |
| Bondville | BON (1,2) | | | | X | X | X | X | X | X | X | X | X | X | X | X | X | X | X | X | X | X | X | X | X | X | X |
| Boulder, SURFRAD | BOS (1,2) | | | | X | X | X | X | X | X | X | X | X | X | X | X | X | X | X | X | X | X | X | X | X | X | X |
| Boulder | BOU (1) | X | X | X | X | X | X | X | X | X | X | X | X | X | X | X | X | X | X | X | X | X | X | X | | | |
| Brasilia | BRB | | | | | | | | | | | | | | | X | X | X | X | X | X | X | X | X | X | X | |
| Cabauw | CAB (1) | | | | | | | | | | | | | X | X | X | X | X | X | X | X | X | X | X | X | X | X |
| Camborne | CAM | | | | | | | | | | X | X | X | X | X | X | X | X | X | X | X | X | X | X | X | X | X |
| Carpentras | CAR | | | | | X | X | X | X | X | X | X | X | X | X | X | X | X | X | X | X | X | X | X | X | X | X |
| Chesapeake Light | CLH | | | | | | | | | | X | X | X | X | X | X | X | X | X | X | X | X | X | X | X | X | X |
| Cener | CNR | | | | | | | | | | | | | | | | X | X | X | X | X | X | X | X | X | X | X |
| Cocos Island | COC | | | | | | | | | | | | | | X | X | X | X | X | X | X | X | X | X | X | | |
| De Aar | DAA | | | | | | | | | X | X | X | X | X | X | | | | | | | | | | X | X | X |
| Darwin | DAR (1,2) | | | | | | | | | | X | X | X | X | X | X | X | X | X | X | X | X | X | X | | | |
| Desert Rock | DRA (1,2) | | | | | | | X | X | X | X | X | X | X | X | X | X | X | X | X | X | X | X | X | X | X | X |
| Concordia Station | DOM (1) | | | | | | | | | | | | | | | X | X | X | X | X | X | X | X | X | | | |
| Darwin Met Office | DWN | | | | | | | | | | | | | | | X | X | X | X | X | X | X | X | X | | | |
| Eureka | EUR | | | | | | | | | | | | | | | X | X | X | X | X | | | | | | | |
| South. Great Plains | E13 (1,2) | | | X | X | X | X | X | X | X | X | X | X | X | X | X | X | X | X | X | X | X | X | X | | | |
| Florianopolis | FLO | | | X | X | X | X | X | X | X | X | X | X | X | | | | | | | | | X | X | X | X | X |
| Fort Peck | FPE (1,2) | | | | X | X | X | X | X | X | X | X | X | X | X | X | X | X | X | X | X | X | X | X | X | X | X |
| Fukuoka | FUA | | | | | | | | | | | | | | | | | | | X | X | X | X | X | X | X | X |
| Goodwin Creek | GCR (1,2) | | | | X | X | X | X | X | X | X | X | X | X | X | X | X | X | X | X | X | X | X | X | X | X | X |
| Gobabeb | GOB (1) | | | | | | | | | | | | | | | | | | | | | X | X | X | X | X | X |
| Neumayer Station | GVN (1) | X | X | X | X | X | X | X | X | X | X | X | X | X | X | X | X | X | X | X | X | X | X | X | X | X | X |
| Ilorin | ILO | X | X | X | X | X | X | X | X | X | X | X | X | X | X | | | | | | | | | | | | |
| Ishigakijima | ISH | | | | | | | | | | | | | | | | | | X | X | X | X | X | X | X | X | X |
| Izana | IZA | | | | | | | | | | | | | | | | | | X | X | X | X | X | X | X | X | X |
| Kwajalein | KWA | X | X | X | X | X | X | X | X | X | X | X | X | X | X | X | X | X | X | X | X | X | X | X | X | X | X |
| Lauder | LAU | | | | | | | | | X | X | X | X | X | X | X | X | X | X | X | X | X | X | X | | | |
| Lerwick | LER | | | | | | | | | | X | X | X | X | X | X | X | X | X | X | X | X | X | X | X | X | X |
| Lindenberg | LIN | | | X | X | X | X | X | X | X | X | X | X | X | X | X | X | X | X | X | X | X | X | X | X | X | X |
| Langley Res. Center | LRC | | | | | | | | | | | | | | | | | | | | | | | X | X | X | X |
| Momote | MAN | | | | | X | X | X | X | X | X | X | X | X | X | X | X | X | X | X | X | X | X | | | | |
| Minamitorishima | MNM | | | | | | | | | | | | | | | | | | | X | X | X | X | X | X | X | X |
| Nauru Island | NAU | | | | | | | | X | X | X | X | X | X | X | X | X | X | X | X | X | X | X | | | | |
| Ny-Alesund | NYA (1) | X | X | X | X | X | X | X | X | X | X | X | X | X | X | X | X | X | X | X | X | X | X | X | X | X | X |
| Palaiseau | PAL | | | | | | | | | | | | | | | X | X | X | X | X | X | X | X | X | X | X | X |
| Payerne | PAY (1) | X | X | X | X | X | X | X | X | X | X | X | X | X | X | X | X | X | X | X | X | X | X | X | | | |
| Rock Springs | PSU (1,2) | | | | | | | X | X | X | X | X | X | X | X | X | X | X | X | X | X | X | X | X | X | X | X |
| Petrolina | PTR | | | | | | | | | | | | | | | X | X | X | X | X | X | X | X | X | | | |
| Regina | REG | | | | X | X | X | X | X | X | X | X | X | X | X | X | X | X | X | X | X | X | | | | | |
| Rolim de Moura | RLM | | | | | | | | | | | | | | | | X | | | | | | | | | | |
| Sapporo | SAP | | | | | | | | | | | | | | | | | | | X | X | X | X | X | X | X | X |
| Sede Boqer | SBO | | | | | | | | | | | | X | X | X | X | X | X | X | X | X | X | | | | | |
| Sao Martinho | SMS | | | | | | | | | | | | | | | X | X | X | X | X | X | X | X | X | X | | |
| Sonnblick | SON | | | | | | | | | | | | | | | | | | | | | | X | X | X | X | X |
| Solar Village | SOV | | | | | | | X | X | X | X | X | | | | | | | | | | | | | | | |
| South Pole | SPO (1) | X | X | X | X | X | X | X | X | X | X | X | X | X | X | X | X | X | X | X | X | X | X | X | X | X | X |
| Sioux Falls | SXF (1,2) | | | | | | | | | | | | X | X | X | X | X | X | X | X | X | X | X | X | X | X | X |
| Syowa | SYO (1) | | | X | X | X | X | X | X | X | X | X | X | X | X | X | X | X | X | X | X | X | X | X | X | X | X |
| Tamanrasset | TAM | | | | | | | | | X | X | X | X | X | X | X | X | X | X | X | X | X | X | X | X | X | X |
| Tateno | TAT (1) | | | | | X | X | X | X | X | X | X | X | X | X | X | X | X | X | X | X | X | X | X | X | X | X |
| Tiksi | TIK (1) | | | | | | | | | | | | | | | | | | | X | X | X | X | X | X | X | X |
| Toravere | TOR (1) | | | | | | | | X | X | X | X | X | X | X | X | X | X | X | X | X | X | X | X | X | X | X |
| Xianghe | XIA | | | | | | | | | | | | | | X | X | X | X | X | X | X | X | X | X | X | | |