# Peer review of "Baseline Surface Radiation Network (BSRN): structure and data description (1992-2017)"

_Earth System Science Data, 2018_

## Referee Comment (RC1) · M. Wild (Referee) · 20 Mar 2018

Review of "Baseline Surface Radiation Network (BSRN): structure and data description (1992-2017)" by Amelie Driemel et al.

This is a nice overview and update on the current status of BSRN. The manuscript provides a detailed description of the available stations, parameters and data, as well as the data formats and associated analysis tools, which should provide a useful documentation for any user of the BSRN data. The manuscript is well written and easy to read. I personally would have highlighted the scientific impacts of BSRN more. But I understand the authors want to focus on the technical and data documentation aspects of BSRN (as the title indicates), and that is probably ok for the purpose of this

manuscript.

After 20 years since the last peer-reviewed publication of BSRN, I believe that BSRN really deserves an update in the peer-reviewed literature, and I wholeheartedly support its publication.

Specific comments:

P2 L7: replace "with data" by "delivering data to the archive"

P2 L8: "distributed over all continents": this excludes stations in oceanic environments. Although very limited in number in BSRN, they should still be referred to as well.

P2 L15: Add "e.g.," in front of the references, as there are many other studies that point this out.

P2 L23: "temporal resolution"

P2 L29: "see also" at wrong position

P2 L33: "for use in satellite and climate model validation"

P3 L4: Add Ohmura et al. 1998 to this reference list

P3 L11: see comment P2 L7

P3 L16: either use consistently shortwave/longwave, short-wave/long-wave, or short wave/long wave throughout, but not in a mixture as it is now in the manuscript.

P4 L5ff: How about aerosol information?

P4 L14: maybe "types of instruments"

P5 L13: expand what Solpos means.

p5 L28: "corrections can be made using algorithms that make use of the other co-located data that is part of the BSRN instrument suite". What other data are these?

p6 L4/5: "Other interesting papers on the quality and possible biases of radiation mea-surements were published by Vuilleumier et al. (2014), Olefs et al. (2016) and Nyeki et al. (2017)". It would be helpful for a reader to expand a bit on these to get an idea whether they are of relevance for his/her application. Just "interesting" is a bit vague.

p6 L11: "LR0100+0300 or LR0100 and LR3010" , add here "see Table 1" to clarify the meaning of this statement.

p7 L11: "Arithmetic averages can be calculated via "Method" but the results should be considered with care in cases where significant data gaps exist." I think this is a critical point, it is good that it is mentioned here, but I would emphasize this even more strongly. I have seen several applications where BSRN monthly means have been generated this way without any critical assessment. I would even opt to remove this function from the data warehouse as the risk of misuse is very high, as also past experience showed.

p7 L25ff: As mentioned in the general comment. I personally would have highlighted the scientific impacts of BSRN more, and not just put it in one small paragraph as part of the summary. But as said I understand the focus of the manuscript is on the technical aspects of BSRN, so I do not request to expand this further.

p7 L26: The first study that made full use of the BSRN network to estimate trends and constituted the "solar brightening" was: Wild, M., Gilgen, H., Roesch, A., Ohmura, A., Long, C., Dutton, E., Forgan, B., Kallis, A., Russak, V., and Tsvetkov, A., 2005: From dimming to brightening: Decadal changes in solar radiation at the Earth's surface. Science, 308, 847-850, which could also be mentioned here.

p8 L5: I think the focus should not be simply on "as many stations as possible", but rather on the worldwide coverage including particularly also ocean and remote land areas, as well as on the coverage of all major climate regimes.

p9 L17: Should the complete reference not read: König-Langlo G, Sieger

R, Schmithusen H, Bucker A, Richter F, Dutton EG (2013) The baseline surface radiation network and its world radiation monitoring centre at the Alfred Wegener Institute. GCOS Report 174: update of the technical plan for BSRN data management. World Meteorological Organization (WMO). http://www.wmo.int/pages/prog/gcos/Publications/gcos-174.pdf

Figure 1: I would also add the yellow circle, indicating stations that measure both upward and downward radiative fluxes, to the legend underneath the map for completeness, as on a first sight one wonders what this prominent yellow circles mean and then first searches for its explanation in the legend. But why only the stations that measure both upward and downward radiative fluxes are labelled with their BSRN abbreviation? This looks a bit like a "two class BSRN society" too me. I think this should be avoided.

Figure 3: what data are underlying the panel to the left? I assume minute data? And from which station over what period of time?

On a personal note it was with great sadness that I leaned on this way that Rainer Sieger passed away. Although I never met him personally I very much appreciated the great support I obtained from him in the process of data submission to PANGAEA.

Martin Wild, 20. 3. 2018
* * *

---

## Author Comment (AC1) · 16 Apr 2018

The authors sincerely thank Mr. Wild for his kind words and his valuable comments, which we all incorporated and used to significantly improve the manuscript.

———

Details below (comment ==> our answer)

————-

P2 L7: replace "with data" by "delivering data to the archive" ==> has been replaced

P2 L8: "distributed over all continents": this excludes stations in oceanic environments. Although very limited in number in BSRN, they should still be referred to as well. ==>

you are absolutely right, we changed the sentence to "over all continents and oceanic environments"

P2 L13: Add "e.g.," in front of the references, as there are many other studies that point this out. ==> has been added

P2 L23: "temporal resolution" ==> was changed

P2 L29: "see also" at wrong position ==> "see also" was removed as it is not necessary

P2 L33: "for use in satellite and climate model validation" ==> was changed

P3 L4: Add Ohmura et al. 1998 to this reference list ==> reference was added

P3 L11: see comment P2 L7 ==> has been replaced

P3 L16: either use consistently shortwave/longwave, short-wave/long-wave, or short wave/long wave throughout, but not in a mixture as it is now in the manuscript. ==> thanks for this hint, we now use consistently long-wave and short-wave (apart from the reference titles which of course may deviate

P4 L5ff: How about aerosol information? ==> up to now BSRN does not offer aerosol information, this is the focus of other networks

P4 L14: maybe "types of instruments" ==> has been changed

P5 L13: expand what Solpos means. ==> We added in brackets the full name "Solar Position and Intensity"

p5 L28: "corrections can be made using algorithms that make use of the other collo-cated data that is part of the BSRN instrument suite". What other data are these? => The sentence was changed to: ".."...or in other cases corrections can be made us-ing algorithms that make use of co-located pyrgeometer data that is part of the BSRN instrument suite (Dutton et al., 2001; Long et al., 2001; Younkin and Long, 2004)."

p6 L4/5: "Other interesting papers on the quality and possible biases of radiation measurements were published by Vuilleumier et al. (2014), Olefs et al. (2016) and Nyeki et al. (2017)". It would be helpful for a reader to expand a bit on these to get an idea whether they are of relevance for his/her application. Just "interesting" is a bit vague. ==> we added for each a short sentence on the main points of the papers

p6 L11: "LR0100+0300 or LR0100 and LR3010", add here "see Table 1" to clarify the meaning of this statement. ==> has been added

p7 L11: "Arithmetic averages can be calculated via "Method" but the results should be considered with care in cases where significant data gaps exist." I think this is a critical point, it is good that it is mentioned here, but I would emphasize this even more strongly. I have seen several applications where BSRN monthly means have been generated this way without any critical assessment. I would even opt to remove this function from the data warehouse as the risk of misuse is very high, as also past experience showed. ==> we added the sentence "or should best not be used at all" to emphasise that even more. The Warehouse of PANGAEA unfortunately will not remove this functionality just for BSRN

p7 L25ff: As mentioned in the general comment. I personally would have highlighted the scientific impacts of BSRN more, and not just put it in one small paragraph as part of the summary. But as said I understand the focus of the manuscript is on the technical aspects of BSRN, so I do not request to expand this further. ==> thanks, this will maybe be the content of the next BSRN paper

p7 L26: The first study that made full use of the BSRN network to estimate trends and constituted the "solar brightening" was: Wild, M., Gilgen, H., Roesch, A., Ohmura, A., Long, C., Dutton, E., Forgan, B., Kallis, A., Russak, V., and Tsvetkov, A., 2005: From dimming to brightening: Decadal changes in solar radiation at the Earth's surface. Science, 308, 847-850, which could also be mentioned here. ==> thanks for the hint, this paper definitely needed to be included and was therefore added to the list of references using BSRN data.

p8 L5: I think the focus should not be simply on "as many stations as possible", but rather on the worldwide coverage including particularly also ocean and remote land areas, as well as on the coverage of all major climate regimes. ==> you are of course right, we added "representative" in front of "stations". The reason we wrote this is that it becomes increasingly hard to handle everything, as none of the people within BSRN get paid for working for BSRN (it is all "on top" – even the WRMC director is only part-time responsible for the WRMC) and there is no extra money available for handling many more stations or for travel expanses to visit candidate stations.

p9 L17: Should the complete reference not read: König-Langlo G, Sieger R, Schmithusen H, Bucker A, Richter F, Dutton EG (2013) The baseline surface radiation network and its world radiation monitoring centre at the Alfred Wegener Institute. GCOS Report 174: update of the technical plan for BSRN data management. World Meteorological Organization (WMO). http://www.wmo.int/pages/prog/gcos/Publications/gcos-174.pdf ==> good catch, we adapted the reference, BUT as we knew that the pdf link does not work anymore we created and provided a different identifier for the publication (http://hdl.handle.net/10013/epic.42596.d001)

Figure 1: I would also add the yellow circle, indicating stations that measure both upward and downward radiative fluxes, to the legend underneath the map for completeness, as on a first sight one wonders what this prominent yellow circles mean and then first searches for its explanation in the legend. But why only the stations that measure both upward and downward radiative fluxes are labelled with their BSRN abbreviation? This looks a bit like a "two class BSRN society" too me. I think this should be avoided. ==> you are absolutely right (both comments), we only added the labels of the stations with full budget for better readability, but will now add all labels as this was not a very good approach.

Figure 3: what data are underlying the panel to the left? I assume minute data? And from which station over what period of time? ==> yes you assume right, these are
minute data, the station is Fukuoka (Japan) and it contains data from one month. We opted not to add more information in the figure caption as the focus lies on the theory/method here.

---

## Referee Comment (RC2) · Anonymous Referee #2 · 11 Jul 2018

Review ESSD-2018-08 BSRN

Agree with the authors about importance of this data for satellite validation, climate modeling and long-term monitoring of radiation fluxes at the ground.  Good international effort.  Several comments and suggestions below but overall a good product for ESSD.

Many comments below relate to the dispersed nature of quality control, e..g to staff scientists associated with nearly 60 sites, and to the relatively hands-off practice of BSRN with respect to data quality flags, etc.  Possibly unique to BSRN, related to the volunteer aspect, and perhaps fundamentally an asset of the network and the data, but different in substantial ways to other global data monitoring efforts (e.g. mention of ARGO, below).  Perhaps users of station radiation data don't care, but should the authors - particularly given their experience within AWI and with Pangaea - comment on the overall strengths and weaknesses of this network model?

Data downloads easily from Pangaea.  Random sample from TOR (1999, Toravere Estonia) looks clean, well-documented and easy to use.  Many users will need to rename the files from .tab to .tsv for easy use; okay for a few but I would not want to do this for 60 stations and multiple years/files.  Authors have a suggestion?

Page 2, line 29:  "McArthur, 2005, see also):"  'see also' leads nowhere?

Page 3, line 15, radiometers measure at 1 Hz but data average up to 1 minute, for valid reasons. Do the data sources have a standard procedure for this, depending on response time of radiometers as operated?  BSRN doesn't specify local procedures, so long as they receive 1 minute data?

Page 4, line 22: "2 for a visualisation of this data file."  Technically, Figure 2 shows year of operation of all stations, with note on downward only or downward and upward.  But not information on the file structure?

Page 4, line 24:  This sentence implies that each station has an individual station scientist attending to quality control, e.g. 59 different station scientists.  But in fact, one person often oversees data from several or many stations?

Page 5, line 1:  In the data file for TOR 1999, I can view good documentation of changes in horizon views, instrument type, instrument calibration (referenced to WRMC procedures) etc., but how would a user know if that file represents a pre-2007 file accepted at AWI without alteration or a more recent replacement file?  Presence or absence of quality control flags could provide a key indicator, but BSRN doesn't archive QC flags, only provides QC tools?  How would a user know if and when the station scientist changed; need to look at subsequent files to see a name change? Perhaps a user can find all this on the Pangaea data viewer rather than trying to extract it from individual files?

Page 5, AWI QC procedures: Does a user know how many station scientists actually use and apply the toolbox?  Does BSRN / AWI know?  After the more-recent AWI checks, do the files go back to the station scientist for correction followed by resubmission, and/or do the data remain in the Pangaea system but identified by QC flags?

Page 5, line 12: "algorithms (Iqbal 1983, Solpos with refraction and Solpos without refraction, Michalsky)": this is supposed to represent a reference to Michalsky or to the names of pull-down position options in the Toolbox?

Page 5, line 28: "nighttime values >-4 W m-2 can just be removed"; I think you mean values more negative than -4 W m-2 (e.g. -5 W m-2) but the phrasing as written seem somewhat confusing?

Page 5, line 30, negative bias also occurs but is not 'evident' rather than 'visible'?

Using the QC Toolbox and the data viewer, users can generate their own quality filters?  This represents both an advantage and a disadvantage.  The advantage arises because that user could focus only daytime data or only on data for a certain sun elevation angle or on only clear

sky max SWD data (e.g. to match clear-sky satellite images).  But, unless that initial user reports QC filter settings, subsequent users can not check those results?  E.g. the possibility arises of researchers extracting slightly different versions from the raw data?  BSRN loses its quality control in these cases?

Page 5, line 31, here again each BSRN user decides how to deal with IR loss to clear nighttime skies but, unless that user clearly documents the assumptions and treatments, BSRN has again lost control of the QC?

Page 6, riming on sensor domes - indeed a problem!  BSRN seems to step back, reference Lanconelli and Matsui?  E.g. a user can not and should not expect that BSRN data will identify and flag this problem; rather each user will need to develop and implement their own identification and correction schemes?

Page 6, registration for data access: a user gets a clear and valid justification here for why BSRN insists on registration, but this seems to violate at least in spirit the fully open access goals of ESSD?

Figure 1: Interesting, useful, Asia remains a serious gap.  In their discussion of the volunteer nature of the BSRN network, and of the necessarily - and much-welcomed - relatively high data quality standards, can the authors identify which factors (instrumentation, long-term operation, or station scientist effort) represent the limiting factor in most cases?  E.g. what would it take to establish a BSRN station on the Tibetan Plateau?  Has somebody estimated, in one of the cited papers perhaps, what we actually need (a global target coverage?) for surface BSRN-quality sites, both number and location?  E.g. the oceanographers have done a network specification for ARGO (e.g. something like 3000 floats with 2 week reporting times covering 60N to 80S on x degree by y degree average spacing in order to properly resolve upper ocean mesoscale features) which they then use as both justification in their proposals and as an operational metric, how close have they come to their desired coverage.  Perhaps in the founding documents for BSRN someone already did a similar estimate, but if so the authors should tell users how close (or not) BSRN has come to initial coverage targets?

Figure 3: The Toolbox "flags values" but the station database does not permanently record those flags?  E.g. each user needs to run these tools?  Could the Pangaea data system support for BSRN a user log, so users could share notes and advice?  No information about which station these data come from?  Deliberately kept anonymous by the authors?

---

## Author Comment (AC2) · 28 Jul 2018

Review ESSD-2018-08 BSRN

R: Data downloads easily from Pangaea. Random sample from TOR (1999, Toravere Estonia) looks clean, well-documented and easy to use. Many users will need to re-name the files from .tab to .tsv for easy use; okay for a few but I would not want to do this for 60 stations and multiple years/files. Authors have a suggestion?

= > The .tab files open in every program which opens txt files. However, if you want to bulk change the file extension there are various ways to do so, e.g. with the command:

ren *.tab *.txt

[Figure]
* * *
Page 2, line 29: "McArthur, 2005, see also):" 'see also' leads nowhere?

= > Has been fixed
* * *
R: Page 3, line 15, radiometers measure at 1 Hz but data average up to 1 minute, for valid reasons. Do the data sources have a standard procedure for this, depending on response time of radiometers as operated? BSRN doesn't specify local procedures, so long as they receive 1 minute data?

= > It is a standard average. The time stamp of the datapoint is supposed to represent the startpoint of the minute average.
* * *
R: Page 4, line 22: "2 for a visualisation of this data file." Technically, Figure 2 shows year of operation of all stations, with note on downward only or downward and upward. But not information on the file structure?

= > I think you mixed Table 2 for Figure 2 here. Table 2 shows what you are describing, Figure 2 is a visualization of a monthly time series
* * *
R: Page 4, line 24: This sentence implies that each station has an individual station scientist attending to quality control, e.g. 59 difierent station scientists. But in fact, one person often oversees data from several or many stations?

= > Very good comment, thanks, we clarified this adding "(some scientists are responsible for more than one station)"
* * *
R: Page 5, line 1: In the data file for TOR 1999, I can view good documentation of

changes in horizon views, instrument type, instrument calibration (referenced to WRMC procedures) etc., but how would a user know if that file represents a pre-2007 file accepted at AWI without alteration or a more recent replacement file? Presence or absence of quality control flags could provide a key indicator, but BSRN doesn't archive QC flags, only provides QC tools? How would a user know if and when the station scientist changed; need to look at subsequent files to see a name change? Perhaps a user can find all this on the Pangaea data viewer rather than trying to extract it from individual files?

= > Thanks, again very good hint. Actually, it is quite easy to recognize an old file: If it is data from pre-2007, and has the version number 1 (in the station to archive file line 1 last number) OR in PANGAEA if it has 2007 as publication year. We added the following sentence to clarify this also to the reader: "Old files can be distinguished from new ones quite easily: If it is data from pre-2007, and has the version number 1 in the station to archive file (line 1 last number) OR if it has 2007 as publication year in PANGAEA it is an old file."

———————————————

R: Page 5, AWI QC procedures: Does a user know how many station scientists actually use and apply the toolbox? Does BSRN / AWI know? After the more-recent AWI checks, do the files go back to the station scientist for correction followed by re-submission, and/or do the data remain in the Pangaea system but identified by QC flags?

= > Not every station scientist uses the BSRN Toolbox, some use other QC which take into consideration various issues related to their environment. There are different things to consider in humid, cold or very hot climate regimes. BUT as every files gets checked by the Data Curator with the help of the Toolbox QCs there is a uniformity in QC that allows direct comparison of the data quality. And yes, suspicious values are sent back to the station scientist for correction, we talk about this in chapter 3.2 but will

add another sentence to clarify this more.
* * *
R: Page 5, line 12: "algorithms (Iqbal 1983, Solpos with refraction and Solpos without refraction, Michalsky)": this is supposed to represent a reference to Michalsky or to the names of pull-down position options in the Toolbox?

= > The first three are options within the Toolbox, the fourth is another one not yet implemented. We added a reference to more detailed information of those tests Iqbal , M.: Introduction to Solar Radiation, Academic Press New York, 1983. Michalsky, J. 1988. The Astronomical Almanac's algorithm for approximate solar position (1950-2050). Solar Energy 40 (3), 227-235.
* * *
R: Page 5, line 28: "nighttime values >-4 W m-2 can just be removed"; I think you mean values more negative than -4 W m-2 (e.g. -5 W m-2) but the phrasing as written seem somewhat confusing?

= > absolutely right, sentence was changed accordingly.
* * *
Page 5, line 30, negative bias also occurs but is not 'evident' rather than 'visible'?

= > has been changed
* * *
R: Using the QC Toolbox and the data viewer, users can generate their own quality filters? This represents both an advantage and a disadvantage. The advantage arises because that user could focus only daytime data or only on data for a certain sun elevation angle or on only clear sky max SWD data (e.g. to match clear-sky satellite images). But, unless that initial user reports QC filter settings, subsequent users can

not check those results? E.g. the possibility arises of researchers extracting slightly diïñĂerent versions from the raw data? BSRN loses its quality control in these cases?

= > The Toolbox does not allow for the selection of "clear sky" etc. It only allows to check the quality of the whole data. Any user that work with the data has to document which steps he takes in the processing of BSRN data.

————————————————

R: Page 5, line 31, here again each BSRN user decides how to deal with IR loss to clear nighttime skies but, unless that user clearly documents the assumptions and treatments, BSRN has again lost control of the QC?

= > here again, the users are responsible for how they further process BSRN data, BSRN offers the complete dataset, but it is up to the interests and focus of the user how he processes that data further

————————————————

R: Page 6, riming on sensor domes - indeed a problem! BSRN seems to step back, reference Lanconelli and Matsui? E.g. a user can not and should not expect that BSRN data will identify and flag this problem; rather each user will need to develop and implement their own identification and correction schemes?

= > all identified riming problems are removed within the dataset, the methodology and equipment for mitigating/reducing contamination of ice in the measurements is the responsibility of the station scientist. There is a chance that some riming events were not detected, but these are the limits of science. BSRN is working to formulate recommendations for the network ————————————————————

R: Page 6, registration for data access: a user gets a clear and valid justification here for why BSRN insists on registration, but this seems to violate at least in spirit the fully open access goals of ESSD?

= > it is indeed open access, as nobody is refused access. We do hope that the reviewer understands the need for pointing out to users the need for correct citation of the data, which still is a big issue.
* * *
R: Figure 1: Interesting, useful, Asia remains a serious gap. In their discussion of the volunteer nature of the BSRN network, and of the necessarily - and much-welcomed - relatively high data quality standards, can the authors identify which factors (instrumentation, long-term operation, or station scientist effort) represent the limiting factor in most cases? E.g. what would it take to establish a BSRN station on the Tibetan Plateau? Has somebody estimated, in one of the cited papers perhaps, what we actually need (a global target coverage?) for surface BSRN-quality sites, both number and location? E.g. the oceanographers have done a network specification for ARGO (e.g. something like 3000 floats with 2 week reporting times covering 60N to 80S on x degree by y degree average spacing in order to properly resolve upper ocean mesoscale features) which they then use as both justification in their proposals and as an operational metric, how close have they come to their desired coverage. Perhaps in the founding documents for BSRN someone already did a similar estimate, but if so the authors should tell users how close (or not) BSRN has come to initial coverage targets?

= > At the BSRN meeting in Boulder (July 2018) we discussed exactly this point, and we are trying to identify gaps e.g. by asking the "satellite community" where they think more ground data is required. However, not only financial, but also on-site operational, and often political issues decide on the feasibility of operating, and maintaining a BSRN station. It also needs to be emphasized that BSRN is a completely volunteer organization, dependent on individual countries and organizations to develop the desire and funding to propose to establish a possible BSRN site. BSRN itself has no funding to establish, operate, or man any sites. Thus while there is desire to "fill in the gaps" there is at present no funding mechanism to do so. There is only the identification of the gaps and informing the community, in the hopes of motivating organizations to establish the

desires long-term, high quality sites traceable to the world SI standards.

—————————————————

R: Figure 3: The Toolbox "flags values" but the station database does not permanently record those flags? E.g. each user needs to run these tools? Could the Pangaea data system support for BSRN a user log, so users could share notes and advice? No information about which station these data come from? Deliberately kept anonymous by the authors?

= > We probably were not clear enough here. The right part is an example of the quality control mechanism within BSRN. These values were found to be suspicious, so the file was sent back to the station scientist and the three days of wrong values (instrument failure) were removed before being archived within the WRMC. We changed the sentence to: "Right: three days of instrument failure shown by data points far below the 1:1 line. These values were reported back to the station scientist and were removed before the file was archived in the WMRC"